# The Uniform Distribution of Hydroxyapatite in a Polyurethane Foam-Based Scaffold (PU/HAp) to Enhance Bone Repair in a Calvarial Defect Model

**DOI:** 10.3390/ijms25126440

**Published:** 2024-06-11

**Authors:** Chiu-Fang Chen, Ya-Shuan Chou, Tzer-Min Lee, Yin-Chih Fu, Shih-Fu Ou, Szu-Hsien Chen, Tien-Ching Lee, Yan-Hsiung Wang

**Affiliations:** 1School of Dentistry, College of Dental Medicine, Kaohsiung Medical University, 100, Shih-Chuan 1st Road, Kaohsiung 80708, Taiwan; lori801107@gmail.com; 2Orthopaedic Research Center, Kaohsiung Medical University, Kaohsiung 80708, Taiwan; yashuanchou@gmail.com (Y.-S.C.); tn916943@gmail.com (T.-C.L.); 3Regenerative Medicine and Cell Therapy Research Center, Kaohsiung Medical University, Kaohsiung 80708, Taiwan; 4Institute of Oral Medicine, National Cheng Kung University, No. 1, University Road, Tainan 701, Taiwan; tmlee@mail.ncku.edu.tw; 5School of Dentistry, National Cheng Kung University, Tainan 701, Taiwan; 6Department of Orthopedics, College of Medicine, Kaohsiung Medical University, Kaohsiung 80708, Taiwan; 7Department of Orthopedics, Kaohsiung Municipal Ta-Tung Hospital, Kaohsiung 80145, Taiwan; 8Department of Orthopedics, Kaohsiung Medical University Hospital, Kaohsiung Medical University, Kaohsiung 80756, Taiwan; 9Department of Mold and Die Engineering, National Kaohsiung University of Science and Technology, Kaohsiung 807, Taiwan; m9203510@nkust.edu.tw; 10Institute of Polymer Science and Engineering, College of Engineering, National Taiwan University, Taipei 106216, Taiwan; samsolvito@gmail.com

**Keywords:** polyurethane, hydroxyapatite, bone grafting, bone regeneration

## Abstract

Polyurethane (PU) is a promising material for addressing challenges in bone grafting. This study was designed to enhance the bone grafting capabilities of PU by integrating hydroxyapatite (HAp), which is known for its osteoconductive and osteoinductive potential. Moreover, a uniform distribution of HAp in the porous structure of PU increased the effectiveness of bone grafts. PEG/APTES-modified scaffolds were prepared through self-foaming reactions. A uniform pore structure was generated during the spontaneous foaming reaction, and HAp was uniformly distributed in the PU structure (PU15HAp and PU30HAp) during foaming. Compared with the PU scaffolds, the HAp-modified PU scaffolds exhibited significantly greater protein absorption. Importantly, the effect of the HAp-modified PU scaffold on bone repair was tested in a rat calvarial defect model. The microstructure of the newly formed bone was analyzed with microcomputed tomography (μ-CT). Bone regeneration at the defect site was significantly greater in the HAp-modified PU scaffold group than in the PU group. This innovative HAp-modified PU scaffold improves current bone graft materials, providing a promising avenue for improved bone regeneration.

## 1. Introduction

Bone grafts are used to heal critical-size segmental defects in bones and address issues in alveolar bone, maxillary sinus augmentation treatment, and maxillofacial skeleton defects. Bone grafts, such as autografts, allografts, or substitutes, are often used to promote bone regeneration. Moreover, these bone grafts are evaluated for their ability to regenerate tissue based on factors such as their osteogenic, osteoconductive, and osteoinductive potential [1]. In the field of bone grafts, autografts are considered the benchmark because they incorporate osteogenic cells and an osteoconductive mineralized extracellular matrix that promotes growth and proliferation. However, their use is hindered by complications such as pain, infection, scarring, blood loss, and donor-site morbidity [2,3,4]. Another option is allografts, which address concerns related to donor site morbidity and limited supply, but have less osteogenic potential than autografts and carry the risk of infection or immune rejection [4,5,6]. Therefore, in recent years, substantial research has emerged to overcome these limitations through bone tissue engineering.

Among possibly useful materials, polyurethane (PU) has great potential. In addition to its nontoxicity, biocompatibility, and biodegradability, the mechanical properties of PU can shift from rigid to flexible by altering its chemical composition, thus expanding its range of applications [7]. Furthermore, these scaffolds create pores suitable for cell growth and enhance their osteogenic, osteoconductive, and osteoinductive potential to mimic the bone microenvironment. Furthermore, PU has also attracted attention because it can be chemically or physically modified [7]. Many studies have modified PU scaffolds for improved bone repair [8,9]. Hydroxyapatite (HAp) has been widely used in tissue engineering and has demonstrated enhanced osteoinductivity, osteoconductivity, high compressive strength, and excellent bone integration [10,11]. More importantly, HAp is the major component of the inorganic phase of bone [12] and is required for bone regeneration [13]. Nevertheless, the use of HAp in bone tissue engineering is limited by its inherent brittleness, presenting challenges in terms of implant molding and biodegradability [14]. In addition, HAp is difficult to mix uniformly with high-viscosity PU materials [14,15], which may affect the physical properties of PU-based bone grafts and inhibit the generation of porous structures.

To address these challenges, this study was designed to develop a HAp-modified PU scaffold with a uniform HAp distribution, osteoinductive properties, and a porous structure tailored for bone graft applications.

## 2. Results

### 2.1. PU and HAp-Modified PU Scaffolds

In this investigation, the prepolymer was synthesized as described in our prior study (Figure 1) [16]. The hydrophobic polyol (Figure 1A) and hydrophilic diisocyanate (Figure 1B) underwent a reaction, yielding a prepolymer featuring three to six isocyanate groups. This prepolymer was further treated with PEG 400 and APTES to serve as an end-capping agent, resulting in the production of a PU foam sponge. In the HAp-modified PU groups, PU was mixed with different ratios of HAp to obtain PU15HAp or PU30HAp (Figure 1C). Additionally, the foaming process of the porous structure was initiated by the interaction between the isocyanate (-NCO) and OH groups, which produced carbon dioxide (CO_2_) gas as a byproduct.

### 2.2. Characteristics of the HAp-Modified PU Scaffold

In the PU foam scaffolds synthesized in this study, the foaming reaction of the porous structure was initiated through the interaction between the isocyanate (-NCO) and OH groups, resulting in the generation of carbon dioxide (CO_2_) gas as a byproduct. Scanning electron microscopy (SEM) images of the scaffolds revealed the morphologies of the pore structures and the distributions of HAp (Figure 2A). The cross-sectional images indicated that all of the scaffolds had microporous structures, and that there were interconnected micropores on the walls of the macropores. The pore sizes of PU, PU15HAp, and PU30HAp ranged from 23 μm to 383 μm, 24 μm to 249 μm, and 24 μm to 333 μm, respectively. Furthermore, the porosities of PU, PU15HAp, and PU30HAp were 85.5 ± 3.2%, 87.2 ± 1.1%, and 86.2 ± 4.6%, respectively (no significant difference). Moreover, the self-foaming reaction and addition of the HAp filler resulted in a uniform distribution of HAp crystals in both PU15HAp and PU30HAp. The HAp particles in our study were rod-shaped, with sizes of 33.7 ± 14.8 μm and 40.6 ± 20.7 μm in PU15HAp and PU30HAp, respectively (no significant difference). Based on the energy-dispersive X-ray spectroscopy (EDS) peaks, we concluded that the crystals in the SEM image were HAp-dispersed by the self-foaming reaction. The distribution of HAp within the PU scaffold was also examined via μ-CT (Figure 2B). The addition of HAp, whether in PU15HAp or PU30HAp, resulted in a highly uniform distribution of HAp. This uniformity was attributed to the CO_2_ byproduct generated during the polymerization process resulting from the reaction between the -NCO groups and water (Figure 1C). The CO_2_ bubbles improved the dispersion of HAp, reducing the likelihood of sedimentation issues during the process.

### 2.3. Chemical Structures of the Scaffolds

The Fourier transform infrared (FT-IR) spectra of PU, PU15HAp, and PU30HAp are shown in Figure 2D. All scaffolds exhibited absorption bands associated with urethane groups, such as C=O at 1600–1700 cm^−1^, N–H at 3400 cm^−1^, and C–O at 750–1150 cm^−1^ [16]. In the FTIR spectra of the scaffolds, the absorption band for –NCO at 2265–2270 cm^−1^ was not observed, which confirmed complete polymerization during synthesis. The FT-IR spectrum did not differ significantly from those of PU, PU15HAp, and PU30HAp, but hydroxyapatite exhibited a significant peak (PO_4_)^3−^ at 1030 cm^−1^.

Thermogravimetric analysis (TGA) was used to measure the mass changes, thermal decompositions, and thermal stabilities of the composite materials to determine the thermal degradation patterns of the PU scaffolds (Figure 2E). The thermal degradation of PU did not exhibit clear distinctions between the soft and hard segments, indicating material integrity and cross-linking and the absence of any additives. The pyrolysis of PU30HAp showed three distinct stages of thermal degradation. The first stage, occurring between 180 °C and 250 °C, was associated with the loss of bound water from the soft segments. The second stage occurred between 250 °C and 480 °C, during which the hard segments were lost, resulting in a plateau in the degradation curve. The final degradation stage left a residual mass of 28.48%, which remained unaltered, indicating the presence of a thermally stable phase, most likely hydroxyapatite.

### 2.4. Mechanical Properties of the Scaffolds

Tensile testing was performed to characterize the elongation and tensile strength of the prepared scaffolds. As shown in Table 1, the tensile strength increased proportionally with increasing amounts of added HAp, increasing from 0.014 MPa in the PU group to 0.028 MPa in the PU15HAp group and 0.053 MPa in the PU30HAp group. These scaffolds can be used as non-load-bearing bone graft materials. However, PU15HAp and PU30HAp reduced the elongation of the foam material, which was likely due to the presence of HAp.

### 2.5. Swelling and Protein Adhesion

To simulate the implantation of a biomaterial in the human body, experiments were conducted to investigate physiological saline uptake (Figure 3A) and protein absorption (Figure 3B). In the physiological saline uptake test, absorption increased the weights of the PU, PU15HAp, and PU30HAp groups, which were 14.88 times, 6.53 times, and 6.18 times greater than their original weights after 10 min, respectively. There were no significant differences in liquid absorption between the PU15HAp and PU30HAp groups, while compared to the PU group, the addition of HAp reduced water absorption. However, the HAp-modified PU scaffolds still exhibited water absorption, as did the bone scaffold made of foam material.

Protein adsorption by biomaterials is a crucial step in bone regeneration. To assess protein adhesion to the bone scaffold, three groups of scaffolds were exposed to HRP-IgG, and the quantities of adsorbed proteins are shown in Figure 3B. The PU15HAp and PU30HAp groups exhibited significantly greater levels of adsorbed HRP-IgG than the PU group did. Furthermore, there were no statistically significant differences between the PU15HAp and PU30HAp groups. The results suggest that HAp-modified PU scaffolds have great potential as biocompatible materials for biomedical applications.

### 2.6. Cytotoxicities of the Scaffolds

Biocompatibility is a crucial criterion for medical devices. Therefore, we assessed the cytotoxicity of the PU scaffolds with Cell Counting Kit-8 (CCK-8) assays. As shown in Figure 3C, NIH 3T3 fibroblasts treated with the leach liquor extracted from the PU, PU15HAp, and PU30HAp foams exhibited greater cell viability at 24 and 48 h than the negative control group (SDS group) did. Additionally, there were no significant differences observed among the PU, PU15HAp, and PU30HAp groups at 24 and 48 h, and there were no significant differences compared with the control group, which did not have any extracted liquor added. These results suggest that all the groups had negligible cytotoxicity.

### 2.7. Bone Healing Evaluation

To investigate whether HAp-modified PU scaffolds can serve as bone grafts to promote the healing of bone defects, we used a nonunion calvarial defect (5 mm diameter, Figure 4A,B) model in rats. Our previous research confirmed that for patients in whom cranial bone defects were present without any bone graft, these defects did not heal even at 8 weeks postsurgery [17].

In this study, we assessed bone regeneration at 0, 4, 8, and 12 weeks postimplantation by μ-CT (Figure 4C). The energy settings for the μ-CT were adjusted to provide the best visualization of the calvarial bone, and the μ-CT was performed 0 weeks after scaffold implantation. HAp was not visible at this intensity on the μ-CT images of the three groups at 0 weeks, facilitating the analysis of new bone formation. The μ-CT analysis of the PU group revealed a nonunion of the calvarial defect at 4, 8, and 12 weeks after surgery, with only slight new bone formation on the edge of the defect. At 4 weeks postsurgery, more new bone had formed inside the defect in both the PU15HAp and the PU30HAp groups than in the PU group. Our analysis revealed a statistically significant increase in the rate of bone regeneration in the HAp-modified PU scaffold group compared with that in the PU group (Figure 4D). Although no statistically significant difference was observed between the PU15HAp and PU30HAp groups, this finding still suggests that HAp-modified PU scaffolds enhanced bone healing. Therefore, this material has potential for use as a bone graft.

We also confirmed new bone formation in the calvarial defect via von Kossa staining (Figure 4E). A histomorphological analysis revealed obvious bone healing and bony bridging in the PU15HAp and PU30HAp groups. In the PU group, only the original bone tissue was evident, and no tissue formation was observed at the location of the surgical defect. However, within the PU15HAp and PU30HAp groups, the dense growth of new bone tissue was observed in the central region of the surgical defect. These results confirm that the HAp-modified PU scaffolds enhanced the repair of nonunion cranial bone defects.

## 3. Discussion

A scaffolding material that is nontoxic, biocompatible, osteoinductive, and osteoconductive, and has excellent bone integration, has the potential to help millions of patients worldwide annually. In this study, we have successfully addressed the uniform incorporation of HAp in the PU scaffold, and these HAp-modified PU scaffolds could stimulate bone repair in calvarial defects.

In the late 1990s, PU emerged as a promising biomaterial for bone repair. It was also reported that mineralization could be increased by adding HAp to the PU scaffold [7,18]. In previous studies, PU/HAp composites have been shown to enhance mesenchymal stem cell proliferation and osteogenic differentiation compared to HAp-free PU scaffolds [18,19]. Additionally, modifying the surface of HAp nanoparticles with organic molecules improved the compatibility between the inorganic HAp and the organic polymer [15], thereby enhancing the bone mineral density, osteogenesis, and angiogenesis of PU scaffolds prepared by the salt leaching method in rat calvarial bone defects [20]. However, achieving a uniform distribution of HAp, especially within viscous PU materials, has proven challenging. Various PU foam fabrication techniques have been developed to create porous scaffolds [21,22], including the salt leaching-phase inverse technique [23], physical foaming, and chemical foaming [14]. The addition of HAp during PU foam fabrication has several limitations. The salt leaching-phase inversion technique, which is often favored for its simplicity, typically faces challenges, such as the concurrent leaching of HAp during pore formation, excessive HAp addition, resulting in reduced foam density, uneven HAp distribution, leading to weakened mechanical strength, and limitations in scalability. On the other hand, chemical foaming methods, due to their stringent technical requirements, high residues of chemical blowing agents, and few allergic reactions, have limited applicability in medical devices [22,24]. Furthermore, in cases of incomplete polymerization in highly porous foams combined with HAp incorporation, the mechanical strength decreases, the elasticity decreases, and the pore size distribution becomes uneven. During our synthesis process, we noted that the byproduct, CO_2_, played a crucial role in efficiently dispersing HAp within the PU scaffold. Through systematic observations utilizing SEM and μ-CT, we achieved the successful generation of a consistently uniform distribution of HAp throughout the porous PU scaffold in our material synthesis.

The 3D structure of a bone graft should exhibit well-interconnected pores to facilitate cell infiltration and nutrient exchange [25]. In this study, we investigated the effects of adding HAp to PU scaffolds and whether there were any changes in the characteristics of the material. Even though PU30HAp exhibited greater tensile strength than PU15HAp did, there was no significant difference in new bone formation between the two groups. This indicates that the differences in the mechanical properties of these materials had little impact on bone healing. Moreover, compared to previous studies, all scaffolds in our study had lower tensile strengths than normal cortical bone did (51–151 MPa) [4]. However, they provided a more complete filling of defects and were easy to process. Taken together, these scaffolds are suitable as non-load-bearing bone graft materials, such as for maxillofacial skeletal defects or calvarial defects.

The SEM images revealed that the structure of the scaffold was a matrix with interconnected pores. This system had beneficial effects on cell regeneration and new bone formation. More importantly, the presence of HAp in the PU scaffolds, in both PU15HAp and PU30HAp, facilitated greater protein adsorption compared to the PU scaffolds. The significant increase in protein adsorption on the PU scaffolds might be attributable to the presence of HAp nanoparticles. HAp addition has the potential to alter the features of scaffold surfaces, such as chemical composition, electric charge, and morphology [26]. Therefore, this modification significantly increased the opportunities for interactions with proteins. Due to the necessity for protein adsorption before cells adhere to the scaffold, increased protein adsorption by the scaffold could affect subsequent cell adhesion and behavior.

The calvarial defect model has been applied extensively in basic and applied research, allowing for the evaluation of bone regeneration [27]. The combination of scaffold features, such as surface topography, chemistry, and microstructure (pore size and interconnectivity), may have a significant influence on bone formation [28]. Moreover, an ideal bone graft for critical-sized bone defects necessitates biocompatibility, osteoconductivity, and an osteoinductive environment to promote bone regeneration [29,30]. In this study, there was no significant difference in the pore size distribution among the three types of scaffolds, but the potential for promoting bone growth in the PU15HAp and PU30HAp groups was greater than that in the PU group. These data suggest that the impact of PU15HAp and PU30HAp on cranial defect repair should be related to the addition of HAp. HAp imparted osteoconductive and osteoinductive properties, was uniformly distributed within the PU scaffold, and enhanced protein adsorption. We confirmed that both the PU15HAp and PU30HAp groups exhibited significant bone healing beginning at 4 weeks after surgery compared to the PU group. This indicates that HAp-modified PU scaffolds can enhance bone healing, demonstrating their potential as bone grafts.

This study has limitations. First, the PU scaffolds in this study were not biodegradable. In previous research, nonbiodegradable polymers have been applied in tissue regeneration, requiring long-term structural support, such as orbital reconstruction, facial reconstruction, and rhinoplasty [31]. The HAp-modified PU scaffolds in this study may also have potential for these applications. Second, as the volume of the bone grafts increased, the current formulation controlled the uniformity of HAp at heights ranging from 7 to 10 cm. However, if larger specifications are used in the future, the formulation may need adjustment, and the control of the pore sizes will be necessary. Third, the mechanical strengths of the bone grafts are not thoroughly discussed in this study. In situations where bones grow into scaffolds without degradation, further confirmation of the mechanical strength at different bone defect sites is needed. Therefore, further studies are needed to improve the synthesis of PU; for example, the type of oligodiol, diisocyanate, or chain extender that may control the degradation of PUs.

In conclusion, we achieved a uniform incorporation of HAp in a PU scaffold, and these HAp-modified PU scaffolds stimulated bone repair in calvarial defects. This study demonstrated a two-stage PU polymerization method that incorporated HAp into PU materials. Furthermore, our results indicated that HAp-modified PU scaffolds maintained their intrinsic plasticity, mechanical properties, and biocompatibility, revealing their superior potential for bone regeneration. This makes HAp-modified PU scaffolds well suited for the treatment of alveolar bone defects, maxillary sinus augmentation, and maxillofacial skeleton defects.

## 4. Materials and Methods

### 4.1. Materials

To synthesize the HAp-modified PU scaffold, the materials were prepared as described in our previous study [16]. The hard segment of the raw material, 1,6-disocyanatohexane (HDI), was obtained from Aldrich Chemical Co. (Milwaukee, WI, USA) and dried at 60 °C under a vacuum before use. Trimethylolpropane (TMP), polypropylene glycol 2000 (PPG 2000), and (3-aminopropyl) triethoxysilane (APTES) were obtained from Sigma-Aldrich (Burlington, MA, USA) and dried at 60 °C under a vacuum before use. Polyethylene glycol 400 and 2000 (PEG 400, 2000), procured from E. Merck, India Ltd. (Mumbai, India), were used as the raw materials for the soft segment and were dried at 60 °C under a vacuum before use. HAp powder (CAPTAL^®^ SBM, Pune, India) was supplied by PLASMA BIOTAL Ltd. (Kyoto, Japan) and dried at 60 °C under vacuum before use.

### 4.2. Synthesis of PU and HAp-Modified PU Scaffolds

In step (A), TMP and HDI were combined at a molar ratio of 1:3. The resulting mixture was stirred at 80 °C for 90 min under a nitrogen atmosphere. Subsequently, the triisocyanate intermediate and PPG triol (PPG 2000 triol) were blended at a molar ratio of 1:3. The resulting mixture (prepolymer A) was stirred at 80 °C for 90 min under a nitrogen atmosphere. In step (B), HDI and PEG diol (PEG 2000 diol) were mixed at a molar ratio of 2:1. The resulting mixture (isoprepolymer B) was stirred at 80 °C for 90 min under a nitrogen atmosphere. FT-IR spectroscopy was used to confirm the formation of a hydrophilic diisocyanate. In step (C), the hydrophobic polyol and the hydrophilic diisocyanate were combined in a molar ratio of 1:6 and stirred at 80 °C for 90 min under a nitrogen atmosphere. This prepolymer exhibited a hydrophobic interior and a hydrophilic exterior.

Subsequently, in the PU group, prepolymer A, isopolymer B, PEG 400, water, and APTES were mixed at a molar ratio of 1:1:0.4:0.3:0.3 (PU). In the HAp-modified PU groups, PU and HAp were mixed at a molar ratio of 1:0.15 (PU15HAp) or 1:0.3 (PU30HAp). The resulting mixture was stirred at 20 °C for 10 to 20 s under a nitrogen atmosphere to yield a PEG/APTES-modified PU intermediate. Subsequently, the intermediate PU product was coated on a PU film or PE release paper to produce a PU foam sponge. The PU foam sponge was then shaped and placed into a sterilization bag. Sterilization was accomplished with gamma irradiation, conducted by CHINA BIOTECH Corporation, with a gamma irradiation dose of 25 kGy at a dose rate of 10 kGy/h.

### 4.3. Characterization of PU and HAp-Modified PU Scaffolds by SEM, μ-CT, FT-IR, and TGA

For a more comprehensive examination of the HAp distribution within the PU foam sponges at the ultrastructural level, we conducted three-dimensional (3-D) reconstructions of the specimens via high-resolution microcomputed tomography (μ-CT) with a Skyscan 1076 system (Skyscan NV, Kontich, Belgium). For material validation, the specimens were trimmed to dimensions of 10 × 10 × 3 mm, forming elongated cuboids. The samples were scanned at an isotropic voxel resolution of 9 μm with a 0.5 mm aluminum filter, 45 kV X-ray tube voltage, 130 μA tube electric current, and 640 ms scanning exposure time to enable the visualization of HAp. The 3D images were reconstructed for analysis at a scale of 0–0.02 (NRecon version 1.6.1.7; Skyscan NV, Kontich, Belgium). The porosity and pore size of scaffolds, and the size of HAp particles, were analyzed using ImageJ 1.54g9 software (National Institutes of Health, Bethesda, MD, USA).

To characterize the scaffolds, we used attenuated total reflectance-Fourier transform-infrared (ATR-FT-IR) spectroscopy (SHIMADZU IR Spirit, Kyoto, Japan). The thermal degradation patterns of the scaffolds were determined by thermogravimetric analysis (TGA-55) in a nitrogen (N2) atmosphere, with temperatures ranging from room temperature to 650 °C to a heating rate of 20 °C/min. The analyses involved a gradual increase in temperature, and the weight was plotted against the temperature. After data acquisition, curve smoothing and other operations were executed to establish the inflection points. Scanning electron microscopy (SEM, JSM-6390LV, JEOL, Tokyo, Japan) was used to examine the morphologies of the PU, PU15HAp, and PU30HAp foams [32].

### 4.4. Mechanical Properties of the Scaffolds

The elongations and tensile strengths of the scaffolds were assessed using a tensile testing machine (506PC, Taichung, Taiwan) in accordance with the ASTM standard method D 882-02. Samples measuring 25 mm in length, 100 mm in width, and 3 mm in thickness were securely positioned to a probe. The mechanical analyses were conducted with a stretching speed of 10 mm/min and a preload of 0.5 N to ascertain the maximum load.

### 4.5. Evaluation of Water Absorption

To investigate the water absorption of the PU, PU15HAp, and PU30HAp scaffolds, we immersed the scaffolds in phosphate-buffered saline (PBS, pH 7.4) at 37 °C. To precisely determine the time required for the scaffolds to reach moisture saturation, we measured the weights of the specimens at 2, 4, 6, 8, and 10 min. The water uptake was quantified as the percentage increase in weight relative to the initial weight with the following formula:Water uptake rate (%) = [(W_1_ − W_0_)/W_0_] × 100
where W_0_ and W_1_ are the weights (g) of the entire system at the initial time and after the analysis period, respectively.

### 4.6. Protein Absorbance of the Scaffolds

To quantify nonspecific protein binding to PU, PU15HAp, and PU30HAp, we used horseradish peroxidase (HRP)-conjugated anti-immunoglobulin G (IgG) adsorption. All experiments were performed in triplicate. Specifically, the scaffolds were prepared in disks with diameters of 7 mm and heights of 3 mm. These scaffolds were immersed in 125 μL of 2 μg/mL HRP-IgG solution at 37 °C for 90 min, followed by five rinses with PBS to remove the nonadhered proteins. Afterward, the scaffolds were transferred to 24-well plates, and 250 μL of 3,3′,5,5′-tetramethylbenzidine (TMB) substrate (Thermo Scientific™, Thermo Fisher Scientific Inc., Waltham, MA, USA) was added. After a 5 min incubation, the reaction was stopped with 2 M H_2_SO_4_, resulting in a tangerine-colored solution (i.e., relative protein adsorption), and the absorbance was read at 450 nm [33,34].

### 4.7. Cytotoxicities of the Scaffolds Determined In Vitro with NIH 3T3 Cells

Following ISO10993-5, we examined the cytotoxicities of the PU scaffolds toward NIH 3T3 fibroblasts [35] with a Cell Counting Kit-8 (CCK-8; Dojindo Laboratories, Kumamoto, Japan).

To obtain extracts from the PU, PU15HAp, and PU30HAp scaffolds, leach liquor was produced by immersing sterilized specimens (1.4 cm in diameter) in 1 mL of Dulbecco’s modified Eagle’s medium (DMEM; Invitrogen, Carlsbad, CA, USA) at 37 °C for 24 h.

During the cell toxicity analyses, NIH 3T3 fibroblasts were initially seeded into each well of a 96-well plate and incubated for 24 h at 37 °C. Next, the medium was replaced with 500 μL of leach liquor or the negative control (SDS treatment). After 24 and 48 h, the medium was replaced with 100 μL of fresh medium containing 10 μL of CCK-8 solution, and the cells were then incubated at 37 °C for 2 h. Subsequently, the cell viability was calculated by measuring the optical density at 450 nm with an enzyme-linked immunosorbent assay reader.

### 4.8. Animals

Male Wistar rats were purchased from the National Laboratory Animal Center and fed at the Kaohsiung Medical University (KMU) Laboratory Animal Center for 1 week before any experiments were conducted. All procedures were performed under an approved protocol (no. 12-0226) of the Institutional Animal Care and Use Committee (IACUC 110245) of Kaohsiung Medical University. The use and handling of the animals were performed according to the Guide for the Care and Use of Laboratory Animals of Kaohsiung Medical University.

### 4.9. Critical-Sized Calvarial Defects in Rats

For investigations related to bone healing, critical calvarial defects were prepared as described in our previous study [36]. These experiments included 15 (n = 5/group) 8-week-old male Wistar rats and were conducted to validate the effectiveness of the bone scaffold in bone regeneration. Before surgery, anesthesia was induced via an intraperitoneal injection of Ketamine^®^ (150 mg/kg body weight) and xylazine (50 mg/kg body weight) at a 4:1 ratio. The hair on the head was shaved with an electric razor, and after disinfecting the scalp, a longitudinal incision was made. Subcutaneous tissues were dissected to access the cranial bone. A 5 mm diameter hole was created in the rat cranial bone with a trephine bur and simultaneous irrigation with PBS to avoid thermal damage. Various bone graft (PU, PU15HAp, and PU30HAp) materials were then inserted vertically into the cranial bone defects of the rats. Finally, the incision was sutured with 5–0 nylon sutures. All of the experimental rats survived without any wound-related complications.

### 4.10. Microcomputed Tomography (μ-CT)

We used μ-CT analysis with the Skyscan 1076 system (Skyscan NV, Kontich, Belgium) to obtain both qualitative and quantitative measurements of bone regeneration within the calvarial defects at 0, 4, 8, and 12 weeks after treatment. The living rats were anesthetized with the sagittal axis of the cranium perpendicular to the scanning plane. The animals were scanned through a 360-degree rotation angle at a 35 μm pixel size with a 55 kV X-ray tube voltage, 180 μA tube electric current, and 130 ms scanning exposure time to provide the best visualization of the calvarial bone.

A cylindrical region of interest (ROI) measuring 5 mm in diameter and concentric to the calvarial defect site was selected for analysis based on the CT data set. This ROI covered the original defect and the surrounding calvarial bone region. The bone growth volume was measured as the bone volume per total tissue volume (BV/TV) using CT volume (CTVol; Skyscan, Kontich, Belgium).

### 4.11. Von Kossa Staining

We used von Kossa staining for histological visualizations of calcium deposits within tissue samples procured from the calvarial bones of the rats. The harvested calvarial bones were fixed in 10% buffered formalin for preservation. After fixation, dehydration was achieved with a graded alcohol series. The specimens were then infiltrated and embedded in acrylic resin (Technovit 9100; Kulzer Technik, Frankfurt, Germany), gradually transitioning to pure resin for optimal tissue stability. Once embedded, the hardened resin-embedded samples were meticulously sectioned into thin 5 µm slices with an ultramicrotome and glass knives to ensure precise sectioning. Subsequently, the sections were immersed in a 1% potassium hydroxide solution for pretreatment to enhance the visibility of calcium deposits.

For von Kossa staining, the pretreated sections were exposed to a 10% silver nitrate solution (Sigma-Aldrich, St. Louis, MO, USA) while being exposed to light, resulting in the formation of silver deposits around the calcium. The staining procedure was completed by treating the specimens with 1% potassium oxalate. The stained sections were subjected to microscopic examination to assess the presence and distribution of calcium deposits.

### 4.12. Statistical Analyses

SPSS version 20.0 was used for the statistical analysis. The experimental data are expressed as the mean ± standard deviation. Statistically significant differences between groups were analyzed using a one-way ANOVA with Tukey’s post hoc test for multiple comparisons. A *p* value less than 0.05 was considered to indicate statistical significance.

## Figures and Tables

**Figure 1 ijms-25-06440-f001:**
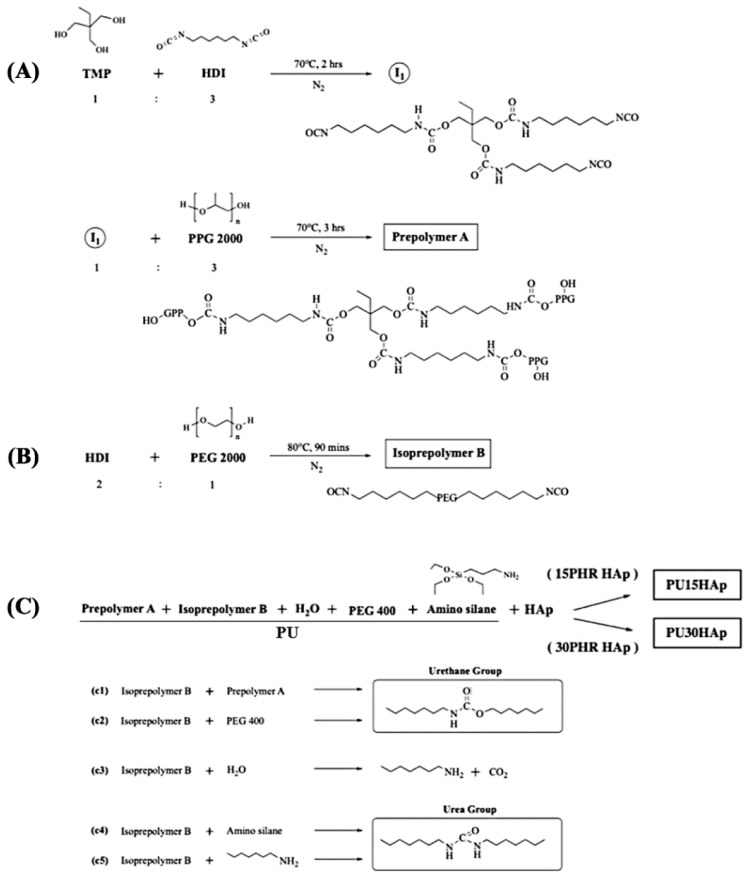
Synthetic routes to 15PHR hydroxyapatite (HAp) and 30PHR HAp-modified polyurethane (PU) bone scaffolds: (**A**) synthesis of the prepolymer A; (**B**) synthesis of isopolymer B; and (**C**) final synthesis of PU15HAp and PU30HAp.

**Figure 2 ijms-25-06440-f002:**
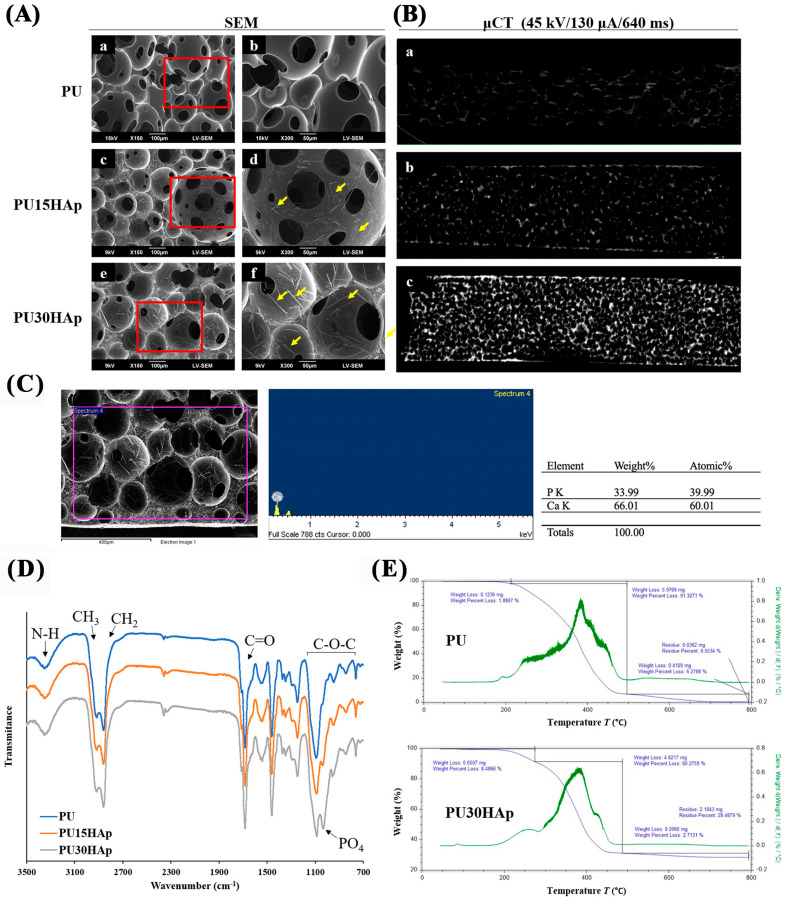
Micromorphologies and chemical structures of HAp-modified PU scaffolds. (**A**) Scanning electron microscopy (SEM) images of the pore structures of the (a,b) PU, (c,d) PU15HAp, and (e,f) PU30HAp scaffolds. (**B**) Three-dimensional reconstruction of μ-CT images of (a) PU, (b) PU15HAp, and (c) PU30HAp. (**C**) The energy-dispersive X-ray spectroscopy (EDS) peaks of HAp in the scaffold. (**D**) Fourier transform infrared (FT-IR) spectra of the PU, PU15HAp, and PU30HAp bone scaffolds. (**E**) Thermogravimetric analysis (TGA) data for PU and PU30HAp.

**Figure 3 ijms-25-06440-f003:**
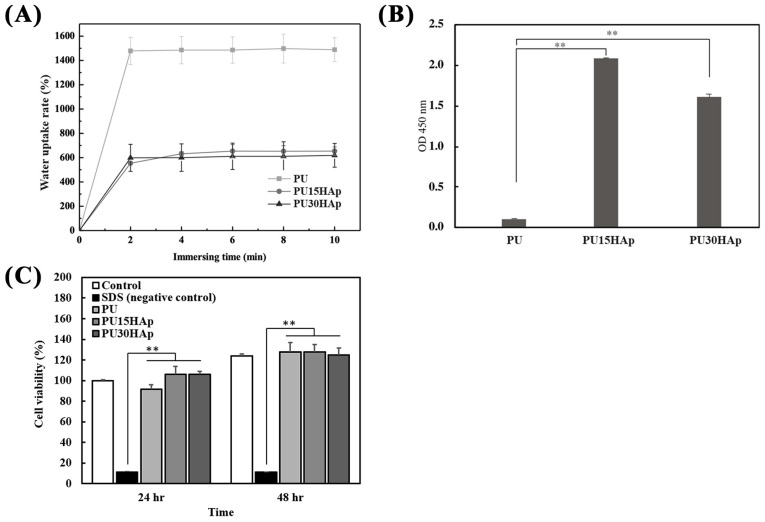
Characteristics of the HAp-modified PU scaffold; (**A**) water uptake rate, (**B**) protein absorption, and (**C**) NIH-3T3 cell viability of PU, PU15HAp, and PU30HAp. ** indicates a significant difference (*p* < 0.01).

**Figure 4 ijms-25-06440-f004:**
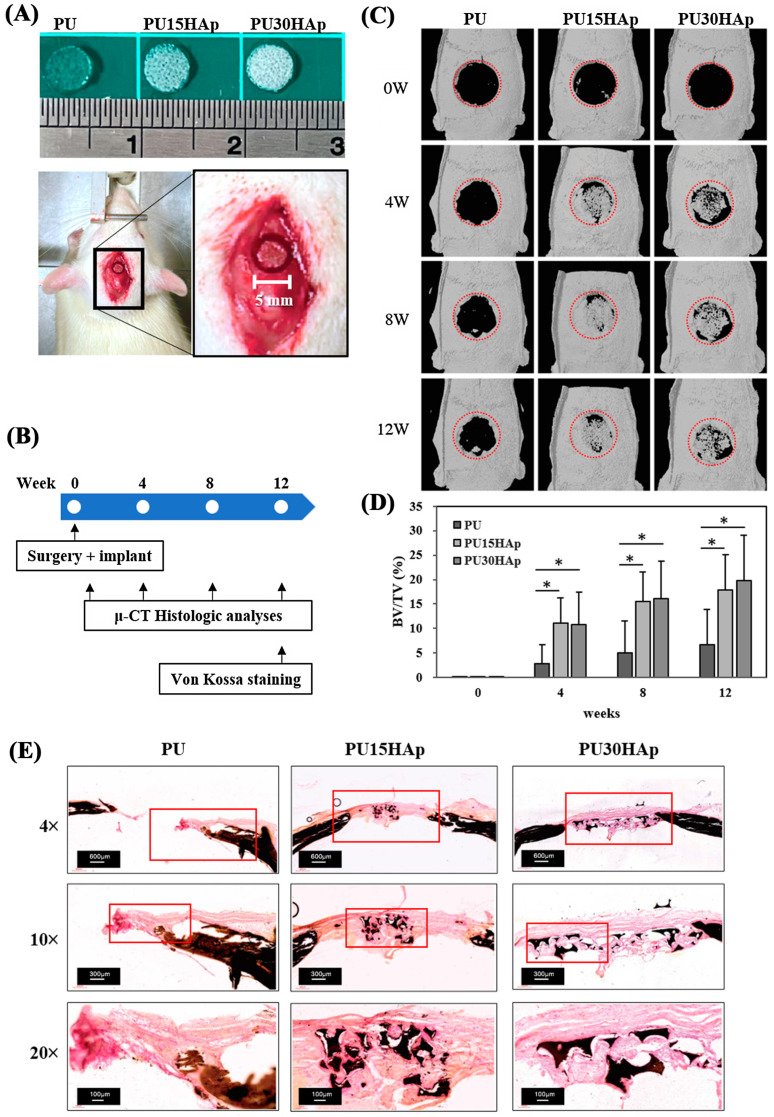
Representative calvarial bone defects in the rat model. (**A**) A 5-mm-diameter defect was implanted with the PU, PU15HAp, or PU30HAp scaffold. (**B**) Timeline for the experimental design of this study. (**C**,**D**) Three-dimensional reconstruction of μ-CT images of bone regeneration in rat calvarial defects at 0, 4, 8, and 12 weeks postimplantation. The red dotted line is the original location of calvarial bone defect. (**E**) Von Kossa staining of PU, PU15HAp, and PU30HAp. * indicates a significant difference (*p* < 0.05).

**Table 1 ijms-25-06440-t001:** Physiological properties of PU, PU15HAp, and PU30HAp.

Group	Tensile Strength (MPa)	Elongation (%)
PU	0.014 ± 0.0005	227.28 ± 0.5757
PU15HAp	0.028 ± 0.0009 *	60.68 ± 0.2983 *
PU30HAp	0.053 ± 0.0005 *^#^	63.97 ± 1.3118 *

Means ± standard errors (*n* = 3). * indicates a significant difference compared with the PU group (*p* < 0.05); ^#^ indicates a significant difference compared with the PU15HAp group (*p* < 0.05).

## Data Availability

All data collected or analyzed during this study are included in this article.

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
