# Peer review of "The Uniform Distribution of Hydroxyapatite in a Polyurethane Foam-Based Scaffold (PU/HAp) to Enhance Bone Repair in a Calvarial Defect Model"

_ijms, 2024, doi:10.3390/ijms25126440_

Round 1

Reviewer 1 Report

Comments and Suggestions for Authors

Introduction

Page 2, line 67: “HAp is difficult to mix uniformly with high-viscosity PU materials…:” please add literature references to support this statement.

Experimental:

Hap provider, physical-chemical properties.

As the assessment of Hap distribution within the scaffolds is a key issue for the proposed work, I suggest to describe in more detail the MicroCT protocol used to evaluate this property. Furthermore, the authors should comment about the capability/resolution of the MicroCT technique to detect the Hap within the scaffolds, with respect to the size and shape of the hap particles (that must be indicated in the manuscript).

Results

Page 3, line 94: “all of the scaffolds had microporous structures with pore sizes ranging from 30 μm 94

to 200 μm…” the size distribution of the different scaffolds should be assessed.

The authors should also provide the overall scaffolds porosity and interconnectivity. All these parameters can be assessed by MicroCT analysis.

Page 3, line 97: EDS X-ray spectroscopy results should be included in the manuscript. If possible, EDS mapping of the cross-section of the scaffolds should be reported to corroborate the uniform distribution of Hap in the scaffolds.

The exposure of the Hap particles to the scaffolds surface should be evaluated as, depending on the materials and process used for scaffolds preparation, the polymer may partially/completely cover the filler.

Page 3, line 100/Page 8, line 220: it is not clear how “the CO2 byproduct produced during the mixing process” may enhance the uniformity of distribution of the HaP in the scaffold.

Water uptake results of Figure 3A suggest that the samples achieved the equilibrium weights before the first set point (day2). I recommend the assessment of the weight change at shorter times to evaluate possible differences with composition.

Cell viability results: The cell viability results are not clear. The authors should explain how it is possible to obtain cell viability higher than 100% after 48 hours. Are the data normalized? Furthermore, in page 6, line 162 the authors stated that “fibroblasts treated with the leach liquor extracted from PU, PU15HAp, and PU30HAp foams exhibited greater cell viabilities at 24 and 72 hours than the control group”. It is not clear what can be the beneficial effect of scaffold extract on cell viability, and how this extract can be better than the cell culture medium. Data at 72 hours are not reported. All these points require a careful revision/explanation.

Please check the manuscript to revise some mistakes, such as:

Page 10, line 302: “The samples were anesthetized and scanned a….”

Page 5, line 150: “Figure 2B” should be “Figure 3B”

The Discussion section does not provide a adequate discussion of the results with respect to the literature on similar systems. The optimization of HaP particles loading into porous scaffolds and the role of these particles on the in vitro and in vivo bone regeneration properties have been widely investigated in the past decades. The authors should clearly explain the novelty of the proposed work before to consider the manuscript for possible publication in International Journal of Molecular Sciences.

Comments on the Quality of English Language

The English must be improved.

Author Response

Dear reviewers,

We would like to thank the reviewers for their suggestion. We have carefully read the suggestions and did some modification in the enclosed revised manuscript. Listed below are the comments/questions posed by the reviewers, along with our replies, including the specific locations of text that were revised in the manuscript. The English grammar and word usage in the manuscript has been checked by a professional editor (American Journal Experts). We hope the manuscript can be published in International Journal of Molecular Sciences after our replies.  Thank you again for the opportunity to revise our manuscript for further review. We hope we have satisfactorily addressed all of the suggestions that arose during the review of our paper.

Sincerely

Yan-Hsiung Wang

Reviewer 2 Report

Comments and Suggestions for Authors

The authors developed a polyurethane scaffold integrated with hydroxyapatite to improve bone grafting outcomes. They used a combination of SEM, FT-IR, and TGAto thoroughly characterize the chemical structure of the hydroxyapatite-modified polyurethane scaffold. Additionally, they examined the scaffold's mechanical properties and cytotoxicity. To evaluate bone regeneration, they conducted micro-CT at various stages, with results indicating that the hydroxyapatite-modified polyurethane scaffolds could facilitate bone healing. I recommend the manuscript for publication in IJMS following minor revisions.

minor questions:

  1. Does the pore size of the scaffold influence the bone healing process? Please discuss.
  2. The effect of the scaffold's mechanical properties on bone healing requires further discussion.
  3. A biodegradation test should be conducted to assess the scaffold's longevity and compatibility in biological environments.
  4. The scaffold's osteoinductivity should be evaluated to confirm its potential for promoting bone growth.

Author Response

(The authors gave the same response as above.)

Round 2

Reviewer 1 Report

Comments and Suggestions for Authors

The authors responded, in part, to the reviewer comments. There is still the incongruence between culture time of figure 3c (48 hours) and the text (72 hours) of the revised manuscript

Author Response

Dear reviewer,

We have corrected the error of manuscript (Page 6, line 171,173). The culture time of cell viability in this study was 24 and 48 hours.